# A Comparison of the Clinical and Radiological Extent of Denosumab (Xgeva^®^) Related Osteonecrosis of the Jaw: A Retrospective Study

**DOI:** 10.3390/jcm10112390

**Published:** 2021-05-28

**Authors:** Zineb Assili, Gilles Dolivet, Julia Salleron, Claire Griffaton-Tallandier, Claire Egloff-Juras, Bérengère Phulpin

**Affiliations:** 1Faculty of Odontology, Lorraine University, 7 Avenue de la Forêt de Haye, 54505 Vandoeuvre les Nancy, France; zinebass@gmail.com (Z.A.); claire.juras@univ-lorraine.fr (C.E.-J.); 2Department of Head and Neck and Dental Surgery, Institut de Cancérologie de Lorraine, 54519 Vadoeuvre-lès-Nancy, France; g.dolivet@nancy.unicancer.fr; 3Cellule Data-Biostatistiques, Institut de Cancérologie de Lorraine, 54519 Vandoeuvre-lès-Nancy, France; j.salleron@nancy.unicancer.fr; 4Cabinet de Radiologie RX125, 125 Rue Saint-Dizier, 54000 Nancy, France; cgriffaton-taillandier@imalo-radiologie.fr

**Keywords:** denosumab, osteonecrosis of the jaw, radiological extent, cone beam computed tomography

## Abstract

Medication-related osteonecrosis of the jaw (MRONJ) is a severe side effect of antiresorptive medication. The aim of this study was to evaluate the incidence of denosumab-related osteonecrosis of the jaw and to compare the clinical and radiological extent of osteonecrosis. A retrospective study of patients who received Xgeva^®^ at the Institut de Cancérologie de Lorraine (ICL) was performed. Patients for whom clinical and radiological (CBCT) data were available were divided into two groups: “exposed” for patients with bone exposure and “fistula” when only a fistula through which the bone could be probed was observed. The difference between clinical and radiological extent was assessed. The *p*-value was set at 0.05, and a total of 246 patients were included. The cumulative incidence of osteonecrosis was 0.9% at 6 months, 7% at 12 months, and 15% from 24 months. The clinical extent of MRONJ was significantly less than their radiological extent: in the “exposed” group, 17 areas (45%) were less extensive clinically than radiologically (*p* < 0.001) and respectively 6 (67%) for the “fistula” group (*p* < 0.031). It would seem that a CBCT is essential to know the real extent of MRONJ. Thus, it would seem interesting to systematically perform a CBCT during the diagnosis of MRONJ, exploring the entire affected dental arch.

## 1. Introduction

Denosumab is a human monoclonal antibody that targets the RANK/RANKL system and inhibits the formation, function, and survival of osteoclasts, thereby reducing bone resorption [1,2]. Xgeva^®^ represents one of the two pharmaceutical specialties of Denosumab. Dosed at 120 mg, it is administered once every 4 weeks to prevent skeletal-related events in patients with bone metastases from solid tumors (pathological fractures, bone pain, spinal cord compression …) [2,3]. It may also be prescribed in adults and skeletally mature adolescents with unresectable giant cell bone tumor or when surgical resection is likely to result in severe morbidity [4].

Its side effect in the oral cavity is the osteonecrosis of the jaw. Medication-related osteonecrosis of the jaw (MRONJ) is the general term that includes necrosis of the jaws related to all antiresorptive medications, comprising bisphosphonates, antiangiogenic medications, but also Denosumab. The definition of MRONJ, as stated in the 2014 American Association of Oral and Maxillofacial Surgeons (AAOMS) position paper, is clinical and based on the presence of exposed bone, or bone that can be probed through an intraoral or extraoral fistula for at least 8 weeks without any history of radiation therapy to the jaws or metastatic disease to the jaws [5]. The classification of MRONJ used in this study was endorsed in 2012 by the Italian Societies of Maxillo-facial Surgery (SICMF) and Oral Pathology and Medicine (SIPMO) [6]. It is a clinico-radiological staging system (comporting 3 stages) (Table 1).

The study presented in this paper focused only on denosumab-related osteonecrosis of the jaw (MRONJ). Most cases of osteonecrosis of the jaw occur following tooth extraction or local trauma from an ill-fitting denture, but they can also occur spontaneously [2,7]. To see the extent of the necrosis, a radiological assessment seems necessary. The accuracy, reliability, and reduced radiation exposure of cone beam computed tomography (CBCT) make it the best choice in this pathology [8]. However, after analysis of the radiologic findings, there is sometimes a discrepancy between the clinical and radiological extent of the osteonecrosis [9].

The aim of this study was to evaluate the incidence of osteonecrosis of the jaw associated with denosumab (Xgeva^®^) and to compare the clinical extent of osteonecrosis with its radiographic extent determined by CBCT.

## 2. Materials and Methods

A monocentric retrospective epidemiological study was carried out at the Institut de Cancérologie de Lorraine (ICL). Written consent from patients was waived because the analysis was based on pre-collected data. This study was approved by our institutional review board and was declared to the Commission Nationale de l’Informatique et des Libertés (CNIL) (declaration 2203860).

This study included all patients who received Xgeva^®^ at ICL between January 2010 and December 2018. Patients were identified by searching the medical records for the keyword “XGEVA”. Patients who received radiation therapy to the jaws and patients referred to the ICL dental office for MRONJ when Xgeva^®^ was not prescribed at ICL were excluded from this study.

All dental consultation reports for these patients were analyzed, and the study focused on patients with MRONJ in whom CBCTs were performed. Several CBCTs may have been performed during patient’s follow-up. CBCTs were taken at the same radiology center and the same radiologist reviewed all CBCTs in the study. For each patient, all available CBCTs and all symptomatic clinical areas were analyzed.

Based on their clinical appearance, the areas were dived into two groups: area with bone exposure, i.e., the “exposed” group and area with fistula not associated with bone exposure i.e., the “fistula” group. The main signs and symptoms studied are the following: pain, signs of infection such as gingival/mucosal inflammation, edema in the oral cavity, trismus, fistula(e), pus exudation, and cellulitis. Radiologically various parameters were investigated: periosteal reaction, bone osteolysis, osteosclerosis, heterogeneous bone condensation, lysis of the cortical, sequestrum formation, absence of bone remineralization after tooth extraction, and the presence of intraosseous air bubbles.

The classification of MRONJ used in this study is established by the Italian Society of Maxillofacial Surgery [6]. Each stage is defined according to the clinical and radiological signs visible on CBCT (Table 1).

To compare the clinical and radiological extent of MRONJ in the two groups, teeth were used as a reference, according to the World Dental Federation scoring system, whether present or not in the dental arch. The value of one tooth was assigned to each retromolar trigone and to each angle of the mandible. In case of fistula, the value of one tooth was also assigned. The clinical extent was the number of teeth with bone exposure identified in patient’s dental consultation reports, and the radiological extent was the number of teeth involved in all bone abnormalities observable on CBCT by the radiologist. Then, the difference between the clinical and radiological extent was calculated: if the difference was positive, the clinical extent was greater than the radiological extent; if the difference was 0, there was no difference between the clinical and radiological extent; if the difference was negative, the clinical extent was less than the radiological extent.

The quantitative parameters were described as median and interquartile range (IQR) or mean and standard deviation. The qualitative parameters were described as frequency and percentage. The normality of the distribution was assessed with the Kolmogorov test. Cumulative incidence was described by the Kaplan–Meier method from Xgeva^®^ induction to MRONJ or last follow-up. Comparison between clinical and radiographic extent was performed with the Wilcoxon test. Statistical significance was set at 0.05. Statistical analyses were performed with SAS software, version 9.2 (SAS Institute Inc., Cary, NC, USA).

## 3. Results

### 3.1. Incidence of Denosumab Related Osteonecrosis of the Jaw (Xgeva^®^)

Between January 2010 and December 2018, 246 patients (155 female and 91 male) were treated with Xgeva^®^ for bone metastases (120 mg Denosumab every 4 weeks) or giant cell bone tumor (120 mg Denosumab every 4 weeks with 2 additional doses of 120 mg in the first month of treatment). The median follow-up time was 16 months (IQR [7,8,9,10,11,12,13,14,15,16,17,18,19,20,21,22,23,24,25,26]). At the last follow-up, 38 patients (24 female and 14 male) developed MRONJ. The cumulative incidence of MRONJ was 0.9% at 6 months, 7% at 12 months, and 15% from 24 months onwards (Figure 1). The mean age of these 38 patients was 63 years (±14). The youngest was 37 years old and the oldest was 90 years old.

### 3.2. A Comparison of the Clinical and Radiological Extent of Osteonecrosis of the Jaw

Out of the 38 patients with MRONJ, 24 (63%) had undergone CBCT. The CBCTs of 21 patients (87.5%) were interpreted. Three patients were deceased and their CBCT (3 CBCT or 12.5%) could not be retrieved. A total of 21 patients were included in the study (Figure 2). Their characteristics are described in Table 2. In three patients (14%), the MRONJ lesions were located in the maxilla, whereas in 18 patients (86%), the lesions were located in the mandible.

Fifteen patients (71%) had undergone CBCT immediately after their first consultation, while for the other six (29%), it was prescribed for the follow-up of their disease. The median time from the clinical examination and the CBCT was 8 days (IQR [5; 29]).

A total of 35 CBCTs were interpreted. The stage of the disease for each CBCT was 1a in 6 patients (17%), 1b in 3 patients (9%), 2a in 21 patients (60%), 2b in 5 patients (14%).

A patient could have one, two, or three clinical areas with bone exposure or fistula. A total of 47 clinical areas were compared to their radiological findings. Thirty-eight areas (81%) were in the “exposed” group and nine areas (19%) were in the “fistula” group (Figure 3).

Clinically, in the “exposed” group, 10 areas (26%) had gingival inflammation, 4 areas (11%) had pus exudation, and 6 areas (16%) had dental mobility associated with bone exposure. In the “fistula” group, five areas (56%) had pus exudation, three areas (33%) had gingival inflammation, and three areas (33%) were painful.

Radiologically, in the “exposed” group, 13 areas (34%) had periosteal reaction, 10 areas (26%) had bone osteolysis, 10 areas (26%) had osteosclerosis, and 8 areas (21%) had bony sequestrum. In the “fistula” group, three areas (33%) had periosteal reaction, three areas (33%) had bony sequestrum, three areas (33%) had no bone remineralization after tooth extractions, and three areas (33%) intraosseous air bubbles. All clinical and radiological signs are described in Table 3.

In the “exposed” group, 17 areas (45%) were less extensive clinically than radiologically, 14 areas (37%) were coextensive clinical and radiological, and 7 areas (18%) were more extensive clinically than radiologically (Figure 4 and Figure 5). The median difference between clinical and radiological extent was −1 (IQR [−4;0]). The maximum difference was −8. The clinical extent of osteonecrosis was significantly less than its radiological extent (*p* < 0.001).

In the “fistula” group, six areas (67%) were less extensive clinically than radiologically and three areas (33%) were coextensive clinically and radiologically. No area was more extensive clinically than radiologically. The median difference was −2 (IQR [−3; 0]). The maximum difference was −13. In this group, the clinical extent of osteonecrosis was significantly less than its radiological extent (*p* < 0.031).

## 4. Discussion

### 4.1. Incidence of Denosumab Related Osteonecrosis of the Jaw (Xgeva^®^)

The majority of patients treated with Xgeva^®^ are women due to a more frequent indication for prescription (metastatic breast cancer), but the incidence rate of MRONJ in this study was the same for men and women (15%). The cumulative incidence of MRONJ was 0.9% at 6 months, 7% at 12 months, and 15% at 24 months. These results are higher than those reported in the international literature, where the incidence rate of MRONJ varies between 0.8% and 2%, sometimes up to 10% [10,11,12,13].

Inter-study comparison is difficult, but several hypotheses may explain this high incidence. First, the small number of patients included in this study. Indeed, it has been shown that as the sample size increases, the incidence of MRONJ decreases [2]. The duration of this study could also influence these results. Most of the studies last 34 to 48 months, whereas here, the follow-up is 8 years (96 months). In the study of Loyson et al. with a duration of 81 months, the rate of MRONJ is also high (10%) [12]. Finally, the aggravating factors vary from one study to another. For example, if the 3-month period of bone healing (for oncological reasons) after oral surgery was not met prior to treatment, despite mucosal healing, tooth extraction was an aggravating rather than a mitigating factor in MRONJ [14].

### 4.2. Clinical Signs

In this study, the mandible was more likely to develop osteonecrosis (86%) than the maxilla (14%). A lot of studies have also reported a trend of higher MRONJ in the mandible possibly due to the presence of terminal vascularization [7,8,9].

Thirty-eight areas (81%) had bone exposure. Most of these areas were asymptomatic. Only 10 areas (26%) had gingival inflammation and 4 areas (11%) had a pus exudation. Nine areas (19%) had a fistula. Half of them had associated pus exudation, one-third had gingival inflammation, and one-third were painful. In some cases, cervicofacial cellulitis, trismus, and/or edema in the oral cavity were observed. These non-specific clinical signs and symptoms are an integral part of the disease and may predict the development of MRONJ. Other clinical features not found in this study have been already observed such as sudden dental mobility, nonhealing post-extraction socket, hypoesthesia or paresthesia of the lips, oro-antral communication, or spontaneous fracture [15]. As these clinical signs and symptoms are non-specific, the vigilance of odontologists must be increased and CBCT is essential for the diagnosis of osteonecrosis. The aim is to improve patient management and not to delay possible treatment.

The classification issued by the Italian Society of Maxillofacial Surgery was preferred. Unlike the AAOMS’ classification, the transition from one stage to another is not based primarily on the clinical picture but on the more specific radiological findings. In this study, 60% of patients had asymptomatic diffuse MRONJ.

### 4.3. Radiographic Findings

In this study, the MRONJ needs to be documented by a CBCT. Thus, 37% of patients (14/38) were not included because only a panoramic radiography (PR) had been taken. This is explained by the fact that initially, in cases of osteochimionecrosis, the PR was the reference and CBCT was not systematically prescribed. Nevertheless, with the rise of the CBCT and the technological advances, more CBCT were prescribed. Indeed, the PR is a limited two-dimensional projection, and the differentiation between necrotic and healthy bone is difficult. Several studies have shown that in case of MRONJ, CBCT was superior to PR in defining the nature and extent of radiological changes. Moreover, the detectability of MRONJ is 96% in CBCT versus 54% in PR [16]. CBCT is more sensitive than PR, particularly with regard to periosteal reaction, bone sequestration, and cortical lysis [17,18]. In the recent study of Santos et al. [19] CBCT is compared with PR in the management decision for MRONJ. As a result, half of the dentists of the study changed their management after analysis of CBCT versus PR. Moreover, in comparison with CT scans, CBCT provides a higher spatial resolution that improves the image quality with lower radiation dosages than regular CT scans [20]. As imaging techniques and knowledge evolved, CBCT is now considered to be the gold standard in dental and maxillofacial sectional imaging [19].

Radiographic findings of MRONJ observed in this study were periosteal reaction, bone osteolysis, bone sclerosis, lack of bone remineralization after tooth extraction, and heterogeneous bone condensation (lytic and sclerotic lesion). Less frequently, sequestration, intraosseous air bubbles, and cortical lysis have been observed. These signs are very similar to those found in the literature [21,22,23] except for intraosseous air bubbles.

On CBCT, the multiple spherical infra-millimeter radiolucent lesions evoke intraosseous air bubbles. It has already been described in patients with dysbaric osteonecrosis but never with MRONJ [24]. This phenomenon may be explained by the production of bacterial gas within the infected and necrotic bone. Radiologists must be watchful for the presence of intraosseous air bubbles on CBCT when osteonecrosis of the jaw is clinically suspected.

The specific radiologic features in denosumab-related osteonecrosis of the jaw are less well described than those of bisphophonate-related osteonecrosis of the jaw [11,25,26]. Although their clinical manifestations are similar, the different mechanisms of action of these two drugs may result in a difference in the radiological characteristics of the necroses they cause. In total, in this study, bone sequestration was observed on CBCT in less than a quarter of the cases. Bone sequestration and cortical lysis appear to be less frequent in patients with osteonecrosis of the jaw with denosumab than with bisphosphonates [22,23].

### 4.4. A Comparison of the Clinical and Radiological Extent of Osteonecrosis of the Jaw

The results of this study confirmed that there was a real discrepancy between clinical and imaging, the radiological extent being significantly larger than the clinical extent (Figure 4 and Figure 5). The definition of MRONJ is purely histological and there are no pathognomonic bone signs of MRONJ. Furthermore, denosumab can induced bone reactions that does not lead to MRONJ. However, bones abnormalities used in this study (periosteal reaction, bone osteolysis, bony sequestrum, osteosclerosis, no bone remineralization after tooth extraction, heterogeneous bone condensation, intraosseous air bubble) are radiological arguments in favor of MRONJ. This was confirmed in patients undergoing longitudinal follow-up (Figure 5).

As far as anyone knows, a single study has compared the clinical and radiological extent of MRONJ and comes to the same conclusion. This study suggests that clinical appearance does not reliably predict osseous changes and emphasizes the importance of detailed radiographic evaluation [26].

In patients with fistula, CBCT is essential for diagnosis of osteonecrosis. For MRONJ, whether or not the patients were symptomatic, the true extent of the pathology cannot be determined clinically, so a CBCT must be prescribed. If the CBCT is prescribed during the first consultation, it will allow the practitioner to follow the evolution of the necrosis (improvement or worsening) but also to measure a possible effectiveness in case of treatment. In addition, the maximum difference in the extent of necrosis was important in the “exposed” and the “fistula’ groups. These results demonstrate that the CBCT prescription cannot limit to the clinical area of interest with a reduced field. The entire dental arch affected by the pathology must be explored to assess the real extent of the lesion and the totality of the bony lesions.

According to the extent of the disease, different treatments exist, ranging from simple conservative treatment to more invasive surgery [5]. The more MRONJ is extended, the more the surgical treatment will be major. The patient’s general health condition must be considered, and the risk/benefit ratio of operation must be evaluated in collaboration with oncologist.

Since this study includes few patients, further multicenter studies are needed concerning oral and radiological management of patient with MRONJ.

## 5. Conclusions

In patients treated with denosumab, practitioners should be vigilant because fistula or bone exposure may indicate underlying bone lesions that may be very extensive. In case of MRONJ, a CBCT exploring the entire dental arch seems necessary. Indeed, in this study, only the performance of a CBCT made it possible to highlight the true extent of the osteonecrosis. Although CBCT appears to be indicated at the time of diagnosis or suspicion of MRONJ, further studies are needed on the optimal frequency of imaging examinations to improve the management of MRONJ.

## Figures and Tables

**Figure 1 jcm-10-02390-f001:**
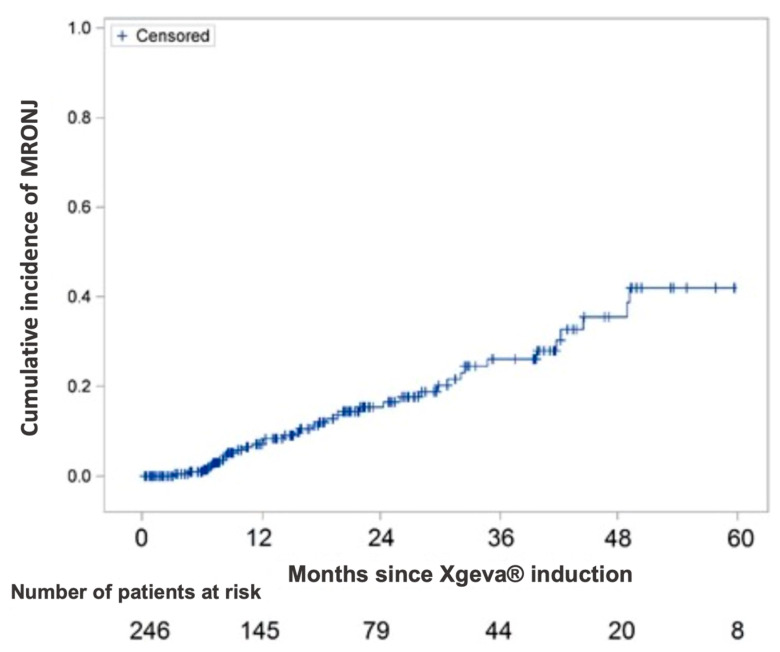
Incidence of MRONJ according to Kaplan–Meier estimator (N = 246).

**Figure 2 jcm-10-02390-f002:**
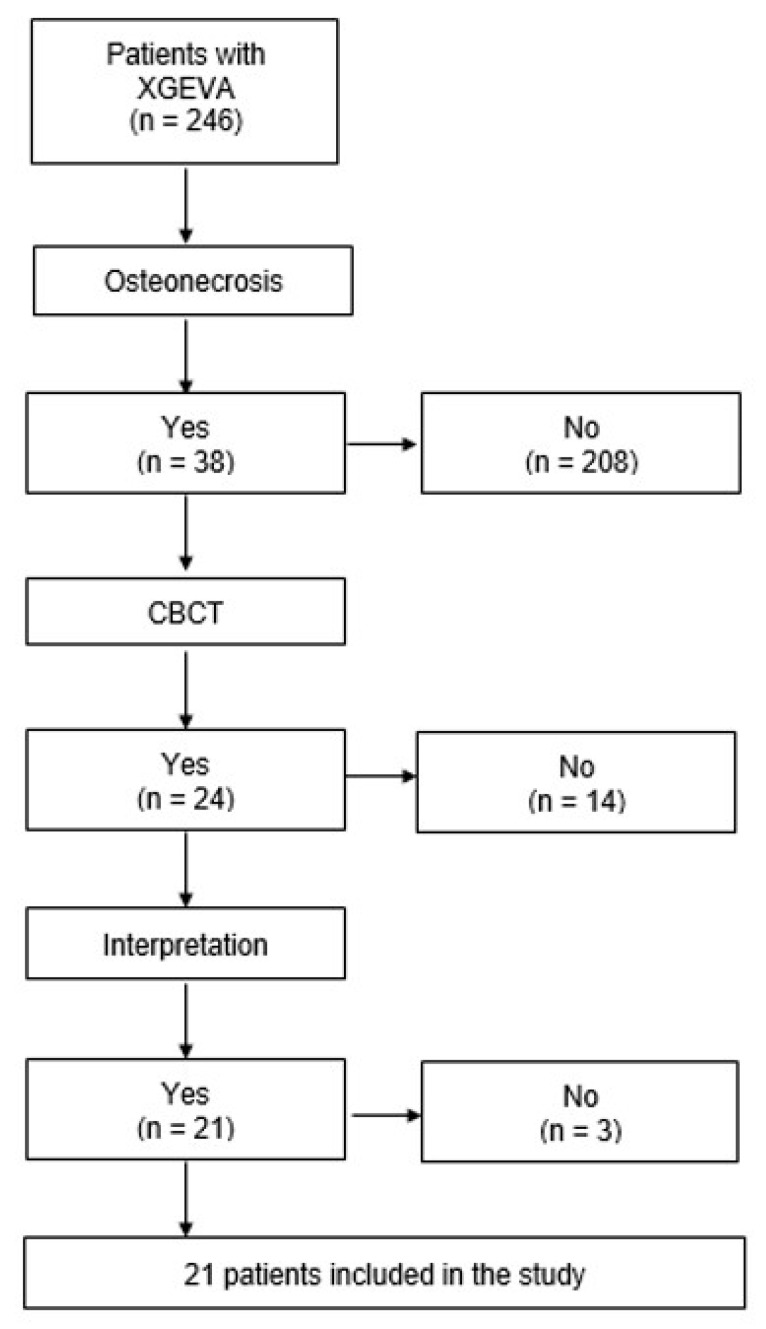
Selection of patients included for the comparison of clinical to radiographic extent assessed by CBCT.

**Figure 3 jcm-10-02390-f003:**
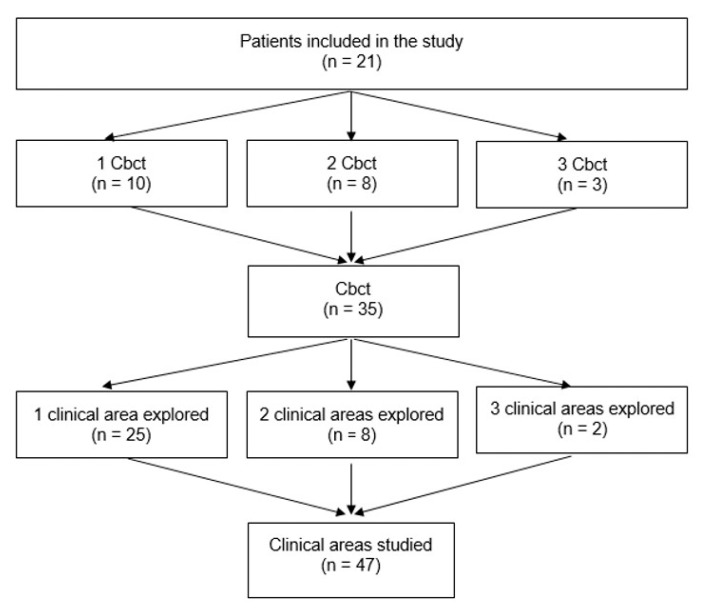
Number of clinical areas studied.

**Figure 4 jcm-10-02390-f004:**
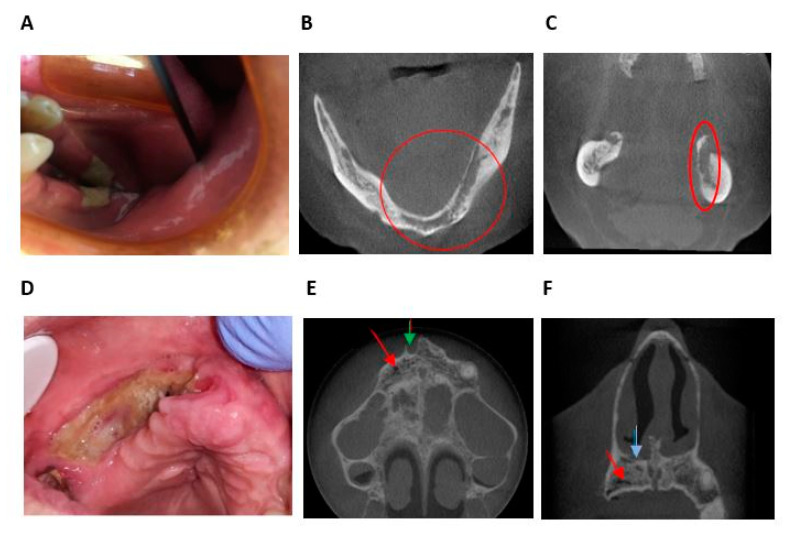
Example of the difference between the clinical extent and radiographic extent of osteonecrosis determined with CBCT. (**A**) Mr X: Intraoral view showing bone exposure 7 mm in the region of the left mandibular molar. (**B**) Mr X: CBCT image of axial section of the mandible, showing bone osteolysis and abnormalities from the incisor region to the left molar region. (**C**) Mr X: CBCT image of coronal section of the mandible showing bone osteolysis and abnormalities of the inner cortical and cancellous bone of the mandible. The radiological extent seems more important than the clinical extent. (**D**) Mrs Y: Intraoral view showing bone exposure with pus exudation from the right incisor region to the right molar region. (**E**) Mrs Y: CBCT image of axial section of maxillary showing dental socket after dental avulsion (green) and intraosseous air bubbles (red). (**F**) Mrs Y: CBCT image of coronal section of the mandible showing intraosseous air bubbles (red) and periosteal reaction (blue) in the right incisor-canine region. The radiological extent seems less important than the clinical extent.

**Figure 5 jcm-10-02390-f005:**
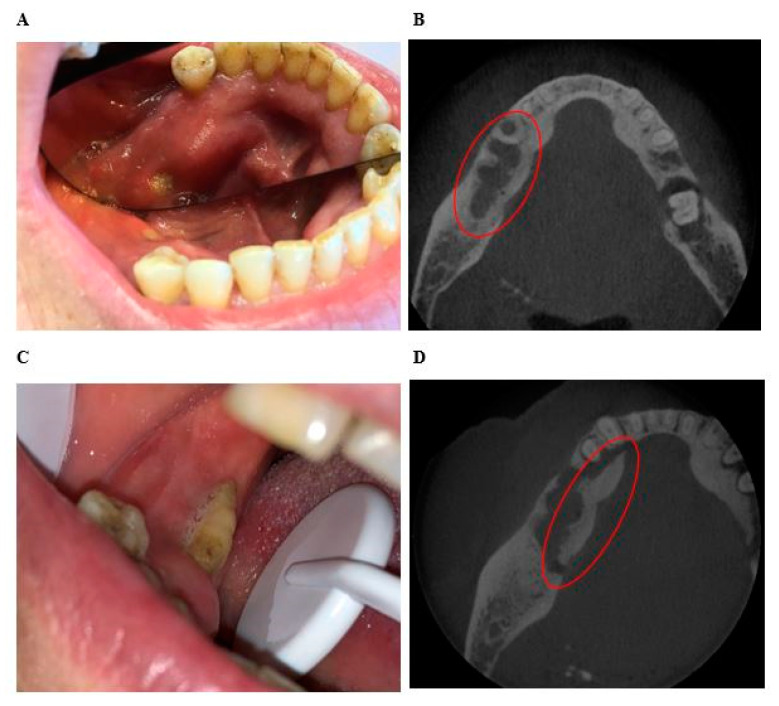
Example of the difference between the clinical extent and radiographic extent of osteonecrosis determined with CBCT in the same patient 7 months apart. (**A**) Mrs C: Intraoral view showing small bone exposure of 3 mm of the inner mandibular cortical of the mandible. (**B**) Mrs C: CBCT image of axial section of the mandible, showing bone osteolysis and abnormalities from the premolar region to the first molar region of the right mandible. The radiological extent is more important than the clinical extent. (**C**) Mrs C: Intraoral view 7 months later showing bone exposure of 8 mm of the crestal and inner cortical of the right mandible (**D**) Mrs C: CBCT image of axial section of mandible 7 months later showing a mandibular sequestration extended from the canine to the first right molar. The radiological extent is more important than the clinical extent.

**Table 1 jcm-10-02390-t001:** Classification of MRONJ by the Italian Societies of Maxillo-facial Surgery (SICMF) and Oral Pathology and Medicine (SIPMO).

Stage	Clinical Signs and Symptoms	Computed Tomography Findings
1: Focal BRONJ 1a. Asymptomatic 1b. Symptomatic (pain and purulent discharge)	- Bone exposure - Sudden dental mobility - Nonhealing postextraction socket - Mucosal fistula - Swelling - Abscess formation - Trismus - Gross mandibular deformity - Hypoesthesia/paraesthesia of the lips	Increased bone density limited to the alveolar bone region (trabecular thickening and/or focal osteosclerosis), with or without the following signs: - Markedly thickened and sclerotic lamina dura - Persisting alveolar socket - Cortical disruption
2: Diffuse BRONJ 2a. Asymptomatic 2b. Symptomatic (pain and purulent discharge)	Same as Stage 1	Increased bone density extended to the basal bone (diffuse osteosclerosis), with or without the following signs: - Prominence of the inferior alveolar nerve canal - Periosteal reaction - Sinusitis - Sequestra formation - Oro-antral fistula
3: Complicated BRONJ	Same as Stage 2, with one or more of the following clinical signs and symptoms: - Extraoral fistula - Displaced mandibular stumps - Nasal leakage of fluids	- Osteosclerosis of adjacent bones (zygoma, hard palate) - Pathologic mandibular fracture - Osteolysis extending to the sinus floor

**Table 2 jcm-10-02390-t002:** Clinical characteristics of 21 patients with MRONJ and available CBCT.

Characteristics		Number (Percentage) or Mean ± Standard Deviation
Cancer localization	Breast	13 (62%)
	Prostate	5 (24%)
	Lung	1 (5%)
	Colon	1 (5%)
	Bone	1 (5%)
Bone metastasis		20 (95%)
Giant cell tumour of bone		1 (5%)
Therapies and risk factors	Diabetes	3 (14%)
	Activ tobacco	1 (5%)
	Alcohol	0 (0%)
	Chemotherapy	13 (62%)
	Hormonotherapy	11 (52%)
	Corticosteroids	6 (29%)
Osteonecrosis	Age of discovery	66 years ±13

**Table 3 jcm-10-02390-t003:** Clinical and radiographic features of MRONJ.

	Description	«Exposed» Group N = 38	«Fistula» Group N = 9	All Areas N = 47
Clinical signs «exposed» group	Gingival inflammation	10 (26%)		
	Suppuration	4 (11%)		
	Bone Mobility	6 (16%)		
Non-specific clinical signs «fistula» group	Gingival/Mucosa fistula		9 (100%)	
	Pus exsudation		5 (56%)	
	Pain		3 (33%)	
	Gingival inflammation		3 (33%)	
	Cellulitis		2 (22%)	
	Trismus		1 (11%)	
	Edema in the oral cavity		1 (11%)	
Radiographic findings	Periosteal reaction	13 (34%)	3 (33%)	16 (34%)
	Bone osteolysis	10 (26%)	2 (22%)	12 (26%)
	Bony sequestrum	8 (21%)	3 (33%)	11 (23%)
	Osteosclerosis	10 (26%)	1 (11%)	11 (23%)
	No bone remineralization after tooth extraction	4 (11%)	3 (33%)	7 (15%)
	Heterogeneous bone condensation	6 (16%)	1 (11%)	7 (15%)
	Sequestrum formation	3 (8%)	1 (11%)	4 (9%)
	Intraosseous air bubbles	1 (3%)	3 (33%)	4 (9%)
	Lysis of the cortical	1 (3%)	1 (11%)	2 (4%)

## Data Availability

The data presented in this study are available on request from the corresponding author. The data are not publicly available, as participants of this study did not agree for their data to be shared publicly.

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
