# Peer review of "A Comparison of the Clinical and Radiological Extent of Denosumab (Xgeva®) Related Osteonecrosis of the Jaw: A Retrospective Study"

_jcm, 2021, doi:10.3390/jcm10112390_

Round 1

Reviewer 1 Report

The manuscript entitled "A comparison of the clinical and radiological extent of Denosumab (Xgeva®) related osteonecrosis of the jaw: a retrospective study" is an interesting retrospective study about the clinical and radiological manifestations of patients affected by DRONJ.

This manuscript may attract the attention of readers, especially given the recent developments on the diagnosis of ONJ.

However, I noted several issues in the manuscript, in particular about the methods.

- Introduction part
In my opinion it must be added the reference about SIPMO-SICMF classification in this part.
AAOMS classification can not be the only classification cited in this part, in particular because the authors used another classification in the text

- Methods part
"All dental consultation reports for these patients were analyzed, and the study focused on patients with DRONJ (as defined by the 2014 AAOMS) in whom CBCTs were performed."

Again the same issue. The AAOMS classification defined as not mandatory the CBCT for diagnosis of MRONJ. 
Please, correct the () part.

FIGURE 4:
In my opinion this picture not clarify the study aim.
The authors MUST report the same prospective of CBCT scans for both cases.
In the FIGURE 4D the authors must report the scan in the transverse plan. Only in this plan you can compare the clinical extension of the bone necrosis with radiological scan.

I also disagree with the authors' statement "Mr X : CBCT image of axial section of the mandible, showing bone osteolysis from the incisor region to the left molar region. The radiological extent is more important than the clinical extent."
In many cases of onj the radiographic findings are not always a sign of the presence of necrotic bone. In the classification used by the authors, for example, there is the definition of focused and diffuse osteosclerosis. However, alterations in bone quality, such as increased medullary trabeculature, do not represent 100% the presence of necrotic bone. In my opinion the extension of the necrosis zone can be best appreciated on CBCT, however not all the radiological signs that are present in patients who develop onj are pathognomonic of the presence of necrotic bone.

The authors must express this concept in the study limitations.

- Discussion part

"In accordance with the definition of the French Agency for the Safety of Health Products and the American Association of Oral and Maxillofacial Surgery (AAOMS), patients had DRONJ manifested by bone exposure or bone that can be probed through a fistula 193 [5]."

This sentence is useless in this section, please delete it

"Other clinical features not found in this study have been already observed such as sudden dental mobility, nonhealing post-extraction socket, hypoesthesia or paresthesia of the lips, oro-antral communication or spontaneous fracture [15]."

Add the following reference that showed other clinical features of ONJ patient [PMID: 33433526]

"According to the extent of the disease, different treatments exist, ranging from simple conservative treatment to more invasive surgery [5]"

Please, add this reference that showed different ONJ management in case of early stages and in accordance to the extension of the disease [PMID: 32615096]

At the end of the manuscript add Abbreviations section.

Reviewer 2 Report

A more detailed description of the Italian classification as compared to the American Staging system as most readers would be more familiar with the American system.

I dont think the term DRONJ is appropriate or useful. Best to use MRONJ

More images would be helpful

Round 2

Reviewer 1 Report

The authors followed the reviewers suggestions improving the content of the manuscript

Author Response

We would like to thank you for your interest in our work.